# Unsupervised Cellular Anomaly Detection in Toxicological Histopathology

**Saketh Juturu**[*]                                          SAKETH.JUTURU@AIRAMATRIX.COM
**Geetank Raipuria**[†]                                  GEETANK.RAIPURIA@AIRAMATRIX.COM
**Raghav Amaravadi**                              RAGHAV.AMARAVADI@AIRAMATRIX.COM
**Aman Srivastava**                                AMAN.SHRIVASTAVA@AIRAMATRIX.COM
**Malini Roy**                                                  MALINI.ROY@AIRAMATRIX.COM
**Nitin Singhal**                                          NITIN.SINGHAL@AIRAMATRIX.COM
**AIRA Matrix, Mumbai, India**

**Editors:** Accepted for publication at MIDL 2025

## Abstract

Irregularities in cellular representation play a crucial role in assessing drug-induced tissue alterations in toxicological histopathology studies. However, the process of annotating rare abnormal cellular variations for training supervised deep learning models presents significant challenges and lacks scalability. While anomaly detection is well-suited for this purpose, it has not yet been explored for cellular-level analysis. In this study, we evaluate cellular anomaly detection using datasets derived from the kidney and liver tissue of Wistar rats. Our findings show that a KNN-distance-based anomaly detection method significantly benefits from employing a feature extractor that has been pre-trained on extensive unsupervised histopathology datasets. When utilizing the best-performing feature extractor, the KNN-distance method surpasses state-of-the-art anomaly detection models by over 4.84% (AUC), including the denoising diffusion probabilistic model, in detecting cellular anomalies. Additionally, we assess the effectiveness of this method in identifying variations in anomalous cell counts between control and treated animal tissues within a toxicological study, revealing a statistically significant difference between the two dosage groups.

**Keywords:** Anomaly Detection, Out-of-distribution Detection, Toxicology, Histopathology, Foundation Models, Cellular Analysis, Drug Safety Assessment.

## 1. Introduction

Toxicological histopathology is essential for non-clinical drug safety evaluations, as it assesses the extent of toxicity induced by a test drug across tissues. It involves analyzing whole slide images (WSI) obtained from laboratory animals that have been exposed to the test drug, with the goal of identifying microscopic tissue changes indicative of toxicity. By comparing the tissue variations in drug-treated animals to those in a control group, researchers can pinpoint abnormal characteristics caused by the drug (Greaves, 2011).

Detecting deviations from normal cell representation is a crucial aspect of a pathologist's routine. For instance, the presence of single cell necrosis in liver tissue and neutrophils in kidney tissue are key indicators. Figure 1, provides samples of normal and abnormal cells in

---

[*] Contributed equally

[†] Contributed equally

liver and kidney tissue. These abnormalities occur in a very small fraction of the tissue and require analysis at high magnifications at which microscopic details of the cell are clearly visible. Consequently, the manual examination of tissue sections for cellular irregularities is labor-intensive and prone to interobserver variability.

Deep learning approaches for cell detection and classification have been widely investigated (Graham et al., 2019; Baumann et al., 2024; Hörst et al., 2024) to aid pathologists in identifying cellular abnormalities. However, generating a large-scale labeled dataset by annotating various cellular anomalies among millions of normal cells is a time-consuming task, even for experienced pathologists. Additionally, while only a few cellular abnormalities are frequently observed, many others are rare.

Anomaly detection (AD) is a vital component of medical image analysis, aimed at identifying deviations from established normal patterns. While there is an abundance of data exhibiting normal characteristics available for training AD models, abnormal data, which encompasses a wide range of variations from the normal, is often scarce or even unknown. AD alleviates the reliance on annotated data and enables the detection of previously unseen variations. This approach is particularly well-suited for preclinical toxicological studies, where unfamiliar representations of cellular variation may arise, making it impractical to train a generalized supervised model.

## 1.1. Related Work

Numerous studies have explored anomaly detection (AD) in medical image analysis (Bao et al., 2024; Cai et al., 2024). Among a variety of anomaly detection methods (Ruff et al., 2021), reconstruction-based, distance-based, or one-class classifier methods have been widely used. Reconstruction-based methods include autoencoders (Baur et al., 2021), Generative Adversarial Networks (GANs) (Goodfellow et al., 2020) and Denoising Diffusion Probabilistic Models (DDPMs) (Ho et al., 2020) that learn to reconstruct normal images. The reconstruction error serves as a scoring function for detecting anomalous samples. Given that the reconstruction model has only seen normal images, a high loss is observed for anomalous samples.

Auto encoders and its variants like Variational AE (VAE) (Kingma et al., 2013), Denoising AE (Kascenas et al., 2022), learn to reconstruct the input image from a low-dimensional latent space representation. GANs employ a generative adversarial approach to learn representations of normal images. For instance, F-AnoGAN (Schlegl et al., 2019) uses a WGAN architecture combined with an additional encoder to map images into latent space for anomaly detection. Another study (Zehnder et al., 2022) incorporates multi-scale input images and perceptual loss to enhance contextual understanding.

DDPMs partially corrupt normal tissue images by adding noise, followed by a denoising the image for a fixed amount of timesteps to reconstruct the image based on the remaining signal. AnoDDPM (Wyatt et al., 2022) suggests using Simplex noise for effective image corruption, while (Bercea et al., 2023) enhances the robustness of diffusion models through the integration of automatic masking, stitching, and resampling techniques. Prior work (Cai et al., 2024; Bercea et al., 2023) found that AutoDDPM outperformed all other reconstruction-based methods.

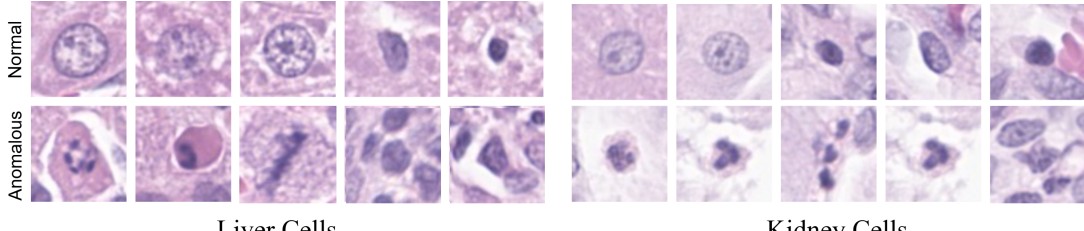

Figure 1: Examples of cells found in the liver and kidney tissues of Wistar rats. Anomalous cells include single cell necrosis, mitosis, and microgranuloma in liver; neutrophils and medullary nephrocalcinosis in kidney. Each patch represents a tissue area of 17x17 micrometers at 40x magnification.

One-class classifier based methods (Zingman et al., 2024; Ruff et al., 2018; Yi and Yoon, 2020; Schölkopf et al., 2001) try to find a hyper-sphere enclosing normal data to identify anomalous samples. Non-parametric K-nearest-neighbor(KNN) distance based approach search within memory bank of normal features to obtain the distance based anomaly score (Sun et al., 2022). Recently, representation discrepancy in teacher-student pair (Salehi et al., 2021; Yamada and Hotta, 2021), and a combination of information from logits and feature embeddings (Wang et al., 2022a) have also been explored. The above methods project data into a learned feature embedding space, to create a better separation between normal and anomalous samples. The feature extractor for these methods is trained on a class-labeled dataset comprising normal in-distribution class(es) (Wang et al., 2022a; Salehi et al., 2021; Li et al., 2023), often through a proxy task such as tissue type classification (Zingman et al., 2024; Dippel et al., 2024), or it may be fine-tuned on the in-distribution dataset (Reiss et al., 2021).

## 1.2. Motivation

Existing approaches often validate their performance using anomalous samples that exhibit significant semantic differences from normal in-distribution data. For instance, tissue necrosis in liver tissue (Zingman et al., 2024) or tumors among benign tissue regions (Cai et al., 2024; Bao et al., 2024; Linmans et al., 2024; Zingman et al., 2024). Such far-out-of-distribution (Far-OOD) samples (Winkens et al., 2020; Linmans et al., 2023) are generally easier to differentiate from normal data. In contrast, as illustrated in Figure 1, the anomalous cells we observe are classified as near-out-of-distribution (Near-OOD). These cells share semantic similarities with normal cells and only exhibit subtle differences. Our experiments indicate that this similarity results in limited performance for models benchmarked for Far-OOD detection. Additionally, many state-of-the-art distance-based methods rely on classifiers trained on labeled class datasets (Wang et al., 2022a; Salehi et al., 2021; Dippel et al., 2024), which poses a challenge when such datasets are unavailable for pre-training. We aim to leverage advancements in foundation models that have been trained on large-scale unsupervised data, which have demonstrated the ability to outperform models trained with supervised data (Caron et al., 2021; Kang et al., 2023; Wölflein et al., 2023). This potential has largely been overlooked in previous research on anomaly detection.

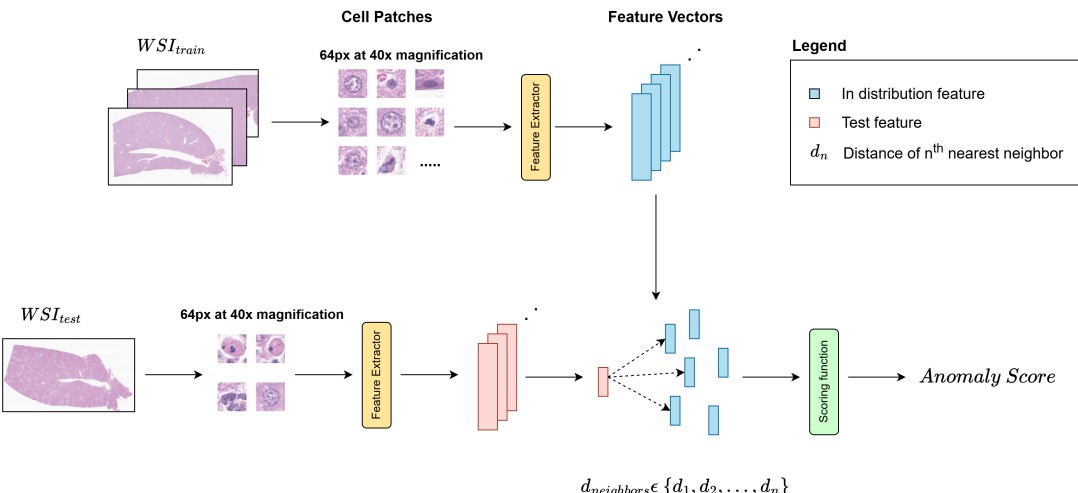

Figure 2: KNN-distance based anomaly detection approach for cellular anomaly detection. A feature extractor trained on large-scale unsupervised histopathology data is employed to obtain feature embeddings from in-distribution data derived from control whole slide images (WSI). The anomaly score for a test patch is determined by calculating the distance to its K-nearest neighbors within the in-distribution feature space.

We introduce a cutting-edge AD method that significantly surpasses existing techniques and establishes a robust baseline for future advancements. Our proposed method calculates the anomaly score based on the distance of a test sample to its K-Nearest Neighbors within the in-distribution feature embedding space, which consists of normal samples. In contrast to previous studies that assessed KNN-distance-based anomaly detection (Reiss et al., 2021; Sun et al., 2022; Linmans et al., 2024), our approach leverages foundation models trained on extensive histopathology datasets to effectively differentiate between ID and OOD samples in the feature embedding space, leading to a notable enhancement in model performance. The key contributions of this work are summarized below:

1. To the best of our knowledge, we are the first to assess a deep learning model for cellular analysis in toxicological histopathology data using unsupervised anomaly detection techniques.

2. We evaluate state-of-the-art foundation models trained on large-scale histopathology datasets for KNN-distance-based unsupervised cellular anomaly detection.

3. We demonstrate that our KNN-distance-based anomaly detection method, when paired with an effective feature extractor, outperforms state-of-the-art anomaly detection models, including diffusion models, in the context of cellular anomaly detection.

4. Finally, in our evaluation of toxicological studies, we demonstrate that the unsupervised method is capable of identifying a higher proportion of cellular abnormalities in drug treated tissues compared to control tissues.

In the following sections, we outline the KNN-distance-based anomaly detection method, followed by a description of the experimental setup and the corresponding results to identify the optimal foundation model for our approach. We will also compare its performance with that of state-of-the-art generative models. Please note that throughout this paper, the terms "anomaly detection" and "out-of-distribution (OOD) detection" are used interchangeably.

## 2. Method

We illustrate our approach via Figure 2 and Algorithm 1, which can be classified as a distance-based method. This method utilizes feature embeddings ($Z_{ind}$) extracted from healthy (training) tissue samples ($D_{in}$) using the feature extractor $ft$, thereby creating a feature space ($D^R$). The anomaly score for a test sample ($x_{test}$) is determined by its proximity to the in-distribution data within this feature space. A cell that closely resembles a healthy cell and is located in a high-density region of the in-distribution feature space will receive a low anomaly score, while a cell that differs from the in-distribution and is found in a low-density region will be assigned a high score. The distance to K-Nearest Neighbors serves as the scoring function. Specifically, we calculate the average distances to K-Nearest Neighbors between the embedding of each test sample and the in-distribution dataset.

---

**Algorithm 1** Anomaly Detection Algorithm

---

**Input**: Normal (training) dataset $D_{in}$, pre-trained feature extractor $ft$, test samples $x_{test}$. For all x $\in D_{in}$, obtain feature vector representations $Z_{ind}$.

**Testing**: Given a test sample $x_{test}$, obtain the feature vector $Z_{test}$ and the k-Nearest Neighbors from $D_{in}$.

**Output**: Anomaly Score based on KNN-distance

---

We use feature extractor pre-trained on large-scale histopathology dataset using self-supervised learning. These foundation models have been shown to surpass the performance of feature extractors trained on supervised dataset, when evaluating for KNN-distance based patch classification, nuclei instance segmentation and image retrieval (Caron et al., 2021; Kang et al., 2023), thus making them effective for our approach. In section 3.2, we compare various state-of-the-art foundation models for cellular anomaly detection.

Our proposed method offer two major benefits over existing methods.

1. **No training required**. The method does not require training on normal in-distribution data. Feature extraction is performed with frozen weights followed by a nearest neighbor search to assign anomaly scores. This significantly reduces the resource requirement for model development. Also, additional in-distribution data can be added at no cost, without model re-training.

2. **The model's performance can be enhanced by improvement in foundation models.** The method exploits feature embedding to identify test samples in low-density regions of the in-distribution dataset, thus, the method's performance can be enhanced with better features that can differentiate normal and anomalous samples. This allows us to benefit from foundation models that are trained on large-scale and diverse unsupervised datasets.

## 3. Experiments

We aim to establish the best method for detecting cellular anomalies by comparing KNN-distance based anomaly detection with state-of-the-art methods. In this section, we first describe the dataset used for evaluation, followed by a comparison of various state-of-the-art foundation models for extracting features in KNN-distance based method, and finally compare the KNN methods with state-of-the-art anomaly detection methods for unsupervised cellular anomaly detection. AUC is used as the performance metric for the evaluation, as used by prior work (Sun et al., 2022; Reiss et al., 2021; Cai et al., 2024; Bao et al., 2024; Fort et al., 2021; Graham et al., 2023). AUC is not a function of a specific threshold on the anomaly score and provides a robust measure of model performance.

### 3.1. Dataset

The dataset used to evaluate cellular anomaly includes a toxicological histopathology study, consisting of WSI from control and high dosaged Wistar Rat, for Liver and Kidney. Specifically, study consists of 14 tissue samples from Control group and 10 samples from drug treated group, for both kidney and liver tissue.

For model evaluation, we created a training and testing cell patch dataset. Specifically, training data is created by annotating cells in fields-of-view with all normal cells, from control WSI. This allows us to create a large pool of in-distribution data which would have near zero abnormal cells. Test set is created by annotating cells on field-of-views from WSI of dosed animal group. Anomalous cell annotations include; Liver: Single Cell Necrosis, Mitosis, Extramedullary Hematopoiesis & Microgranuloma; Kidney : Neutrophils & Medullary Nephro- calcinosis). Table 1 gives an overview of the dataset. An equal amount of normal cells are obtained from dosed animal group. For each annotated cell, a crop of size 64x64px is extracted at 40x magnification, aligning the cell in the center.

### 3.2. Evaluating the best feature extractor

Since performance of KNN-distance based anomaly detection method significantly depends on the feature extractor's ability to segregate normal and anomalous samples in the feature space, we first evaluate with different foundation models (Chen et al., 2022; Kang et al., 2023; Wang et al., 2022b; Filiot et al., 2023; Chen et al., 2024; Zimmermann et al., 2024; Nechaev et al., 2024; Lu et al., 2024), that were trained using varying self-supervised learning techniques and diverse datasets [1]. Table 2 provides results.

---

1. Note, none of the above foundation models have been trained on cell patch data, rather on patches sized 224x224 extracted at different magnifications.

|         | # Liver WSI | # Kidney WSI |
|---------|-------------|--------------|
| Control | 14          | 14           |
| Treated | 10          | 10           |

|       |                 | Liver | Kidney | # WSI |
|-------|-----------------|-------|--------|-------|
| Train | In-Distribution | 1.8M  | 2.2M   | 4     |
| Test  | Normal          | 11496 | 10358  | 10    |
| Test  | Anomalous       | 11941 | 10140  | 10    |

Table 1: The dataset used for performance evaluation. Left: The number of WSI in the toxicological study for each organ. Right: Number of patches and WSI used to create training and testing dataset.

| Method | Model | #WSI | Magnification | Liver | Kidney | Mean |
|---|---|---|---|---|---|---|
| DINO(Caron et al., 2021) | ResNet-50 | NA | NA | 85.06 | 53.02 | 69.04 |
| DINO(Caron et al., 2021) | ViT-S | NA | NA | 82.90 | 51.58 | 67.24 |
| DINO(Caron et al., 2021) | ViT-B | NA | NA | 85.81 | 52.11 | 68.96 |
| HIPT(Chen et al., 2022) | ViT-S | 11K | 20x | 91.95 | 67.66 | 79.80 |
| Lunit(Kang et al., 2023) | ResNet-50 | 21K | 20x,40x | 85.54 | 62.59 | 74.06 |
| Lunit(Kang et al., 2023) | ViT-S | 21K | 20x,40x | 95.50 | 86.71 | 91.10 |
| CPath(Wang et al., 2022b) | ViT-S | 32K | 20x | 89.18 | 84.54 | 86.86 |
| Phikon(Filiot et al., 2023) | ViT-B | 6.1K | 20x | 92.90 | 82.17 | 87.53 |
| CONCH(Lu et al., 2024) | ViT-B | 1.1M* | 20x | 94.25 | 91.71 | 92.98 |
| UNI(Chen et al., 2024) | ViT-L | 100K | 20x | 93.79 | 88.22 | 91.00 |
| Virchow2(Zimmermann et al., 2024) | ViT-H | 3.1M | 5x,10x,20x,40x | **96.97** | **91.83** | **94.40** |

Table 2: Performance comparison of different feature extractor pre-trained on large-scale unsupervised histopathology dataset for cellular anomaly detection using KNN-distance based method. The table reports, model architecture, WSI used, magnification of patches used for training and AUC on anomaly scores of liver and kidney cell patch dataset. Top two scores are highlighted in Bold and Underlined. *CONCH uses 1.1M image text pairs.

| | Liver | Kidney | Mean |
|---|---|---|---|
| f-AnoGAN(Schlegl et al., 2019) | 93.23 | 85.90 | 89.56 |
| AutoDDPM(Bercea et al., 2023) | 82.03 | 75.03 | 78.53 |
| PANDAS(Reiss et al., 2021) | 91.74 | 72.30 | 82.02 |
| KNN with Virchow2 (ours) | **96.97** | **91.83** | **94.4** |

Table 3: Performance comparison of anomaly detection methods on liver and kidney cell patch dataset, comparing KNN-distance based method with state-of-the-art methods. The table report AUC on anomaly scores.

As expected, when comparing performance using ResNet(He et al., 2016) & ViT(Dosovitskiy, 2020) foundation models pre-trained on ImageNet and histopathology datasets, a significant gain in performance is observed when using the domain-specific dataset for pre-training. Next, we compare ResNet(He et al., 2016) and ViT(Dosovitskiy, 2020) architecture, using the weights optimized on the same pre-training histopathology dataset (Kang et al., 2023). Vision Transforms outperforms ResNet, demonstrating that transformers learn better features from large-scale pre-training dataset when compared to the ResNet, as shown in previous works (Caron et al., 2021; Kang et al., 2023).

We observe that the KNN-distance based anomaly detection model's performance improves when using foundation models trained on data that includes samples at 40x magnification, which corresponds to the magnification at which cell patch dataset is extracted for cellular anomaly detection. The model performance shows correlation with an increase in model size and the amount of the pre-training data used, large model size and larger pre-training dataset improve the performance. Virchow2 (Zimmermann et al., 2024) is found to be the best performing feature extractor for anomalous cell patch detection, that uses the largest amount of data for pre-training, extracting tiles at multiple magnifications.

### 3.3. Comparing with state-of-the-art methods

Next, we compare the KNN-distance based method with the state-of-the-art models. Based on previous work (Cai et al., 2024; Bao et al., 2024), we identify three best performing models f-AnoGAN(Schlegl et al., 2019), AutoDDPM(Bercea et al., 2023) and PANDAS(Reiss et al., 2021). F-AnoGAN trains WGAN architecture and an additional encoder using output of the WGAN. The combined anomaly score is computed using a discriminator feature residual error and an image reconstruction error. AutoDDPM, consists of three stages ie. mask, stitch, and re-sampling. Diffusion process is used to generate an initial likelihood map of potential anomalies followed by stitching them with the original image and joint noised distribution re-sampling. Whereas, PANDAS finetunes a pre-trained feature extractor using a compactness loss.

Implementation details for each method are provided in the Appendix section D. Table 3 provides the AUC scores for anomaly detection on liver and kidney tissue cell patches. Our approach, utilizing a self-supervised pre-trained feature extractor and a KNN-distance based scoring function, outperforms other methods for both tissue types. Interestingly, the KNN method using the top four best-performing feature extractors achieves a higher AUC than all three comparison methods. Pre-training on a large-scale dataset enabled our method to achieve superior performance

PANDAS adapts ResNet(He et al., 2016) weights, trained on supervised ImageNet data, to the anomaly detection task using compactness and elastic weight consolidation loss. The AUC score for this method is higher than some of the feature extractors used with the KNN-distance based method, as seen in table 2. Specifically, scores obtained using ImageNet pre-trained weights with SSL significantly underperform compared to PANDAS. However, fine-tuning on an in-distribution dataset falls short when compared to feature extractors trained on larger and diverse histopathology datasets.

Figure 5, in appendix, provides reconstruction of normal and anomalous patches for f-AnoGAN AutoDDPM. We observe that AutoDDPM is able to reconstruct anomalous images with low error, which reduces its ability to identify these images. We believe this can be attributed to subtle variation between normal and anomalous samples, allowing the models to have low reconstruction loss for both image categories. f-AnoGAN(Schlegl et al., 2019) on other hand has lower quality reconstruction for anomalous samples, achieving higher scores than AutoDDPM.

## 4. Evaluation on Toxicological Study

We evaluate the capability of the approach using KNN-distance based unsupervised anomaly method, along with best performing feature extractor - Virchow2(Zimmermann et al., 2024), to detect pathologically relevant changes in the tissue due to the administered drug. Annotating all anomalous cells across 20 WSIs is not feasible due to the high cell count. Thus instead of an AUC score, we analyze cellular distribution for the toxicological study, comparing the number of abnormal cell patches in the control and drug-treated tissue. A higher count of abnormal cells like single cell necrosis, mitosis, and microgranuloma could indicate drug toxicity (Greaves, 2011). We compare anomalous cell count as a percentage of total cell count, to account for tissue area. 10 images from control and drug-treated group each

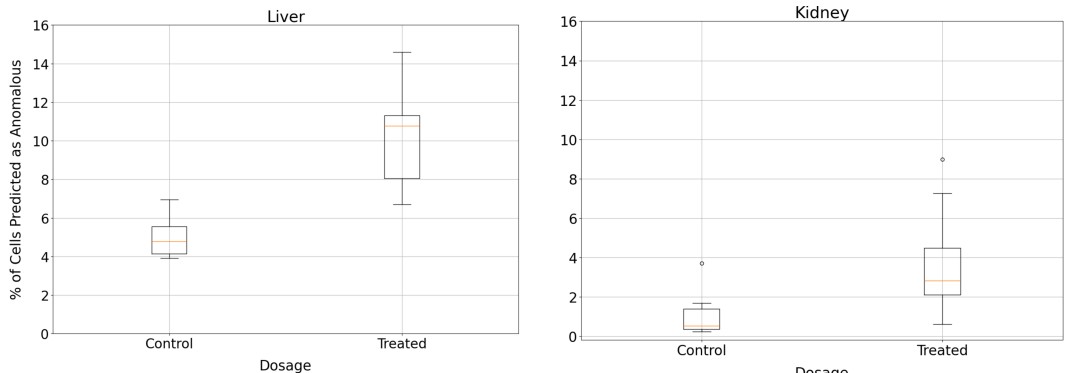

Figure 3: Evaluation of the KNN-distance based unsupervised cellular anomaly detection on a toxicology study, to detect changes in anomalous cell count, on administration of a test drug. The figure shows box plots of percentage of anomalous cells in control and drug-treated tissue. The variation in anomalous cell count obtained a p value of 1.5e-5 & 0.0147 for liver and kidney respectively. Thus, a larger proportion of cells in treated group have higher anomaly score than control group.

are used from kidney and liver, this excludes four WSI used for creating training data to avoid data leakage.

A threshold based on the anomaly scores of in-distribution data is used to classify a cell as abnormal, and is set to $Q_3 + 1.5 \times \text{IQR}$, where Q3 is the third quartile distance & IQR represents Inter-quartile range. Using $Q_3 + 1.5 \times \text{IQR}$ as the threshold allows the rejection of outliers from the in-distribution data. Futher implementation details are provided in appendix section D.5.

Box plot in figure 3 shows the percentage of anomalous cells. A significant increase in percentage of anomalous cells is observed in the drug-treated animal group, A p-value of 1.5e-5 & 0.0147 was obtained liver and kidney tissue respectively. Thus, we can conclude that more cells in the treated group are far from normal cellular representation in the feature embedding space, compared to the control dose group. The compound was confirmed by a pathologist to induce toxicity in liver and kidney tissue, verifying the assessment made using cellular anomaly detection. Figure 4 in appendix provides examples of predictions by the unsupervised anomaly detection algorithm.

## 5. Conclusion

We show that KNN-distance based unsupervised anomaly detection, using vision transformer as a feature extractor, pre-trained on large Histopathology data, achieves high AUC scores for cellular anomaly detection. The method is found to outperform state-of-the-art reconstruction based methods, by exploiting foundation models. The method is found to differentiate between control and drug-treated tissue, based on proportion of anomalous cells, indicating drug toxicity. In the future, we plan to pre-train a feature extractor using large-scale cell patch data from multiple organs, to further improve model performance.

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

## Appendix A. Example predictions of cellular anomaly detection

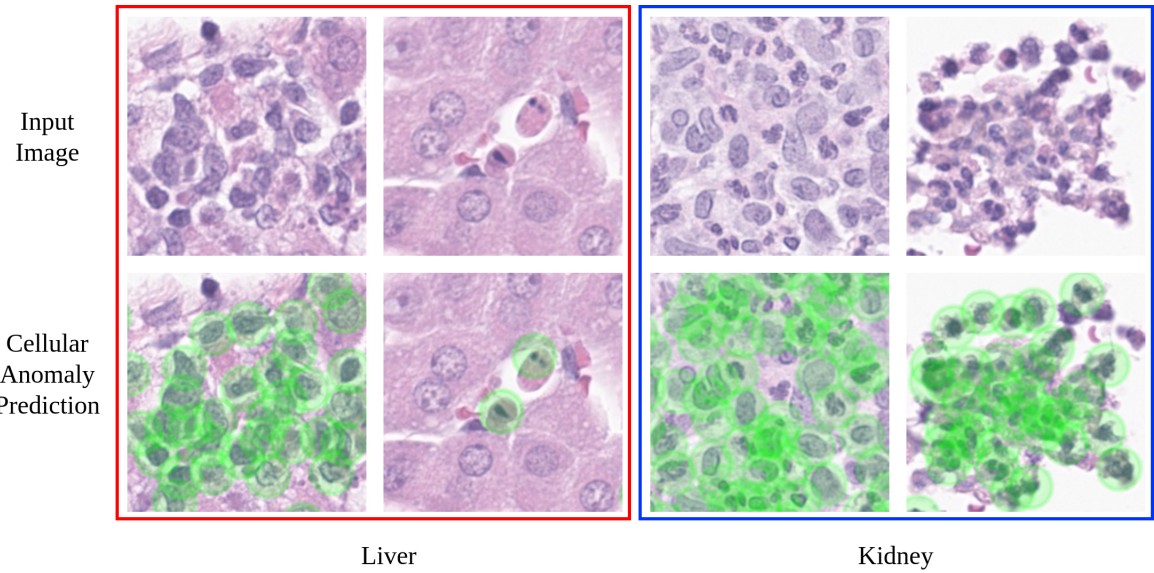

Figure 4: The figure shows example predictions of cellular anomaly detection method using best performing feature extractor and KNN-distance based anomaly score, for liver and kidney tissue. The cells predicted as anomalous are highlighted in green.

## Appendix B. Example cell reconstruction

Figure 5 provides sample reconstruction of normal and abnormal cells for both liver and kidney tissue, obtained from AutoDDPM and f-AnoGAN method. It is observed that AutoDDPM is able to reconstruct the image patch much better than f-AnoGAN, even in case of abnormal samples.

## Appendix C. Benchmarking Cellular anomaly detection based on anomaly type

We created a class-labeled data set to evaluate class-wise performance of all foundation models using KNN-distance based method, and other state-of-the-art methods. For each class - Liver: Single cell necrosis (SCN), microgranuloma (MG) & Extramedullary Hematopoiesis (EMH); Kidney : neutrophils, 2000 cells were identified by the pathologist. Figure 6 provides the class-wise results.

We observed that KNN-distance based method with Virchow2 as feature extractor obtains the highest scores for SCN, MG and Neutrophils classes. However, for EMH class, f-AnoGAN achieves the highest AUC score. LUNIT, CONCH & UNI feature extractors also obtain high class-wise scores.

| Cell Type | Kidney | | | Liver | | |
|-----------|--------|----------|-----------|--------|----------|-----------|
|           | Input  | Auto DDPM | f - Ano GAN | Input | Auto DDPM | f - Ano GAN |
| Normal    |        |          |           |        |          |           |
| Normal    |        |          |           |        |          |           |
| Abnormal  |        |          |           |        |          |           |
| Abnormal  |        |          |           |        |          |           |
| Abnormal  |        |          |           |        |          |           |

Figure 5: The figure shows example reconstruction using f-AnoGAN (Schlegl et al., 2019) and AutoDDPM(Bercea et al., 2023) for normal and abnormal cells. For liver, abnormal cells consists of Single cell necrosis (SCN), (microgranuloma) MG & extramedullary hematopoiesis) EMH, from 3rd to 5th row; for kidney all three abnormal cells are neutrophils.

## Appendix D. Implementation Details

All model were trained and inferred using NVIDIA RTX A4000 GPUs.

### D.1. KNN(Ours)

The code was implemented in Pytorch, and uses Faiss library (Johnson et al., 2019) for the nearest neighbour distance calculation. The anomaly scoring function using KNN-distance based method can be obtained in two ways, using the mean distance of K-nearest neighbours or using the distance to the Kth-nearest neighbor. We find that when using Virchow2 as the feature extractor, the AUC for liver reduced to 96.66% when using distance to the Kth-nearest neighbor, as compared to 96.97%. We also experimented with different k values [10, 25, 50, 100, 200, 500, 1000, 2500] for obtaining KNN-distance and found k=200 to give best results, as seen in figure 7.

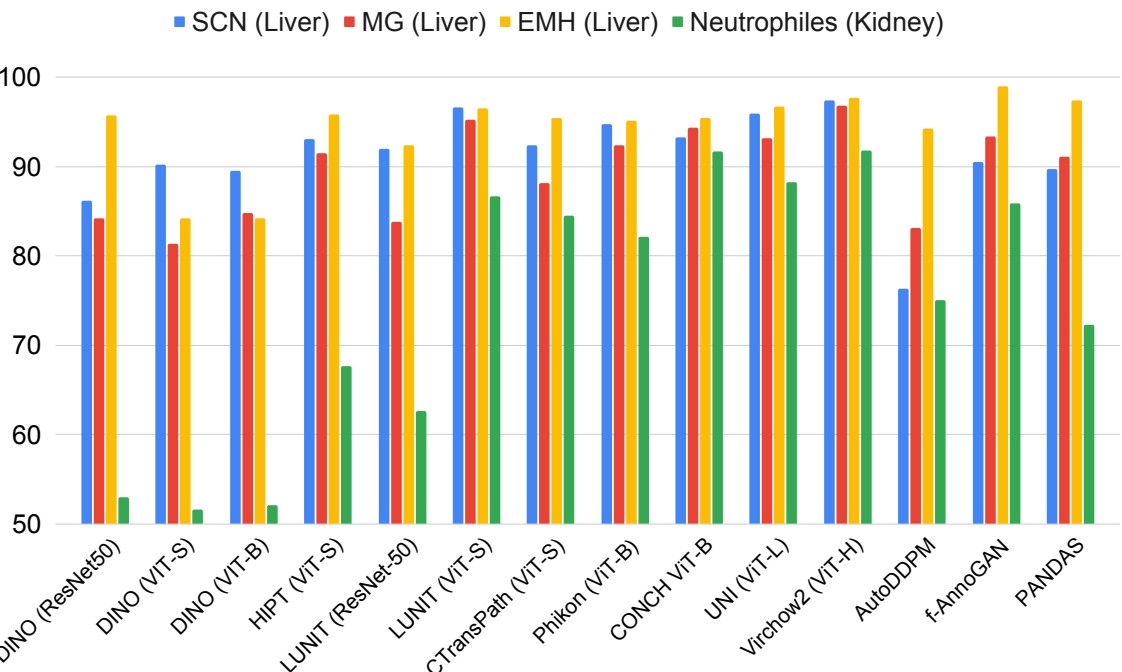

Figure 6: The figure provides class-wise AUC scores for four anomalous cell types, comparing all feature extractors benchmarked for KNN-distance based method and three state-of-the-art methods - AutoDDPM, f-AnoGAN and PANDAS. Virchow2 feature extractor obtains the highest overall results.

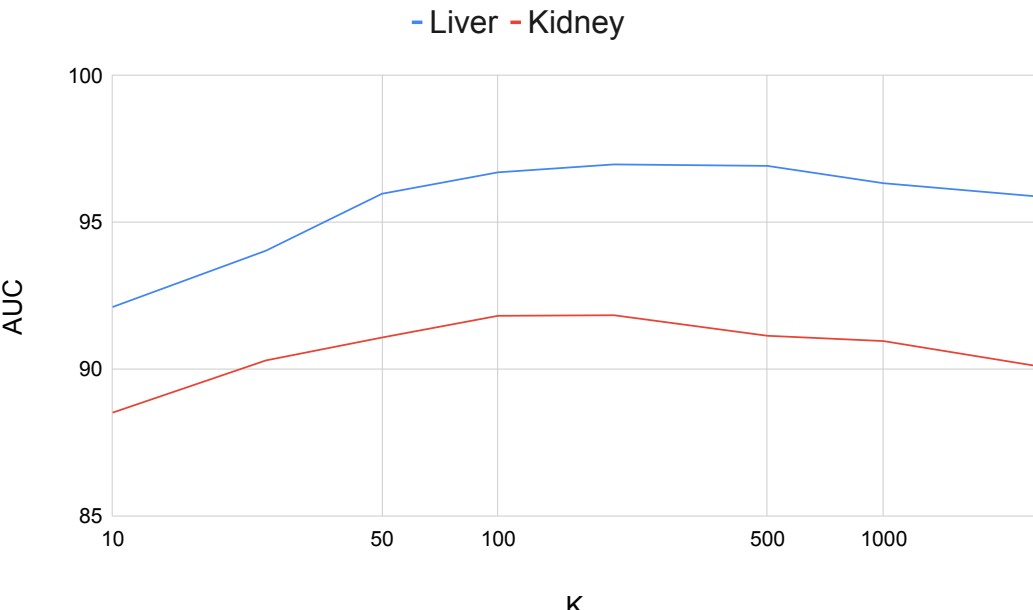

Figure 7: We evaluate the sensitivity of KNN-distance based method to the number of neighbors (K) used to calculate the anomaly score. The figure provides the AUC value for cellular anomaly detection, with Virchow2(Zimmermann et al., 2024) feature extractor, for different K values.

### D.2. AutoDDPM

We use architecture and training procedures as provided (Bercea et al., 2023) using code provided by https://github.com/ci-ber/autoDDPM/tree/main. We use a 3-layer U-Net with [128, 256, 256] channels, one residual block per layer and a single-headed attention block after each residual block with a corresponding spatial dimension of 2. The architecture takes 3 channeled images of size (64*64). The noise level is set to t=200 and resampling steps to 5. We trained two separate models for liver and kidney cell dataset as described in table 1. Both the models were trained for 200,000 iterations using Adam optimizer and Cosine learning rate scheduler with maximum learning rate of 1e-4 and a batch size of 128.

### D.3. F-AnoGAN

We train Wasserstein GAN (WGAN) followed by image-to-image (izi) mapping encoder, as described in (Schlegl et al., 2019), using code available at https://github.com/A03ki/f-AnoGAN. WGAN was trained for 20 epochs with learning rate 0.0002 using ADAM optimizer with the batch size of 32. izi encoder is trained for 20 epochs with learning rate of 0.0002 using adam optimizer with batch size of 128. Combination of MSE loss between original and reconstructed image, and MSE loss between encoder mapping of real and fake image is used as anomaly score, as provided by the github repo.

### D.4. PANDAS

We use best performing feature extractor, Resnet-152 pre-trained on ImageNet dataset, and training setup as described by (Reiss et al., 2021), using code from https://github.com/talreiss/PANDA. The feature extractor is trained for 15 epochs on the training dataset as described in section 1, using a batch size of 1024 with a learning rate of 1e-2. We found K=200 as best performing. Sum of distances of k nearest neighbors of each test feature from the train features is used as the anomaly score.

### D.5. Evaluation on Toxicological Study

For the analysis, cells are detected using pretrained Cell-ViT (Hörst et al., 2024) model, and a crop of size 64x64 px is taken for all the cells, at 40x magnification. To identify anomalous samples in test data, a threshold is applied on the anomaly score. The threshold is based on the anomaly scores of in-distribution data and is set to $Q_3 + 1.5 \times \text{IQR}$, where Q3 is the third quartile distance & IQR represents Inter-quartile range. Using $Q_3 + 1.5 \times \text{IQR}$ as the threshold allows the rejection of outliers from the in-distribution data. Figure 4 provides example field-of-views with anomalous cell prediction. For better visualization, a circular overlay is created around the centroid of the cells.

