# OpenReview forum: "Unsupervised Cellular Anomaly Detection in Toxicological Histopathology"
_MIDL.io/2025/Conference — MIDL 2025 Poster_

### Official Review · Reviewer_SSPF · 2025-02-10

**Confidence:** 4
**Preliminary Rating:** 3
**Recommendation:** Poster
**Final Rating:** 4

**Summary:**

In this paper, the authors propose to perform anomaly detection on a private toxical histopathology dataset (composed of healthy and pathological WSI of rat liver and kidney). To do so they perform feature extraction on WSI patches with large pre-trained ViT or Resnet, and then compute the distance to the K-nearest healthy neighbor to obtain the anomaly score. The author first perform a benchmarking of 9 feature extractors on the task of localizing anomalous patches in the treated rat's kidneys and liver. They then evaluate the best performing feature extractor ability to discriminate between control and treated rat WSI patches.

**Strengths:**

- The proposed method is very simple and thus very simple to understand and to reproduce
- The benchmarking of multiple feature extractors is a useful result for the community
- The performances seem robust to the choice of K (in the KNN algorithm)
- This is the first study that the reviewer found to use KNN with large pre-trained feature extractors for WSI
- No training is required

**Weaknesses:**

- The database used is private
- The methodology is very basic (no true innovation, see for instance Ruff et al 2021 review)
- The framing of the literature review can be improved
- The number of experiments is quite small (2 tasks with the same database)

**Detailed Comments:**

- "Additionally, (Cai et al., 2024; Bercea et al., 2023) found that AutoDDPM outperformed all other reconstruction-based methods." Please introduce briefly AutoDDPM if mentioned in the literature.
- "After training, classifier probabilities (Zingman et al., 2024; Dippel et al., 2024), K- Nearest Neighbor (KNN) distance (Reiss et al., 2021; Sun et al., 2022), or a combination of information from logits and feature embeddings (Wang et al., 2022a) are used as the
scoring function." minor : other methods can be used (GMM, OC-SVM) so I would add that some other methods exist (I feel like the sentence make it seems like only these methods exist).
- In the "toxicological study", the training set was reduced from 14 WSI to 4, does this reduction affect the performances a lot ? The authors could provide the AUC of the best performing feature extractor when using only 4 WSI (as I understand the experiments are pretty fast to run due to the method using no training at all)
- Could the authors motivate the use of another metric than the AUC in the toxicological study ?
- The authors make then comment that their p-value is above 0.05, while giving the exact value. I would suggest removing the statement about the value being under 0.05, as there is a controversy in the broad scientific community about this threshold, see for instance Di Leo, G., Sardanelli, F. Statistical significance: p value, 0.05 threshold, and applications to radiomics—reasons for a conservative approach. Eur Radiol Exp 4, 18 (2020). https://doi.org/10.1186/s41747-020-0145-y.
- I would suggest, to enforce the clarity of the experiments, to use terms like "healthy patches" or "anomalous patches", as it is indeed what's fed to the model and not "cells"
- "We compared using average of K-Nearest Neighbour distance and distance to Kth-nearest neighbor, using the later reduced the performance by 0.31%." Can the authors precise what performance they are talking about ? And maybe give the exact value.


- Typos : "extarctors", "were were", "anomolous", **a** "feature extractor trained on imagenet"

**Justification Of The Final Rating:**

I have read the provided revised manuscript and have decided to change my rating to weak accept, as explained in the comments bellow.
Thanks to the authors for their work and discussion. Thanks to the authors for their work and discussion.

**Justification Of The Preliminary Rating:**

The reviewer is very familiar with the field of anomaly detection (AD) in images, but not so familiar with the histopathology community. After a short literature search, this seems to be the first study to perform AD with pre-trained feature extractors and KNN on WSI histopathology. I would argue that a very basic method, used on a private database, does not justify a publication in the MIDL conference, but maybe I don't realize the interest for the histopathology community of the ViT benchmarking. I will stay very attentive to the reviews of my colleagues and to the response of the authors to adjust my rating.

**Questions To Address In The Rebuttal:**

- The authors often mention that their study is about "cellular-level" analysis or that they perform "cellular anomaly detection", could the authors precise what they mean by "cellular" in the introduction ? Is it relative to the scale at which the images are magnified ? This term might not be obvious for people not working on WSI.
- Minor comment : The authors state that "AD methods can be broadly categorized into two types: reconstruction-based and projection-based.". I would first argue that this dichotomy does not encompass every AD methods, as for instance they are density-estimation or one-class classification methods that do not use projections. For a great review on AD methods see Ruff, L., Kauffmann, J. R., Vandermeulen, R. A., Montavon, G., Samek, W., Kloft, M., ... & Müller, K. R. (2021). A unifying review of deep and shallow anomaly detection. Proceedings of the IEEE, 109(5), 756-795. I would suggest being more nuanced in the classification of the AD methods, as they are numerous ways to classify them.
- "Reconstruction-based methods utilize Generative Adversarial Networks (GANs) (Goodfellow et al., 2020) or Denoising Diffusion Probabilistic Models (DDPMs) (Ho et al., 2020) that learn to reconstruct normal images" : the reviewer feels like this statement is plainly false. See the above review (Ruff 2021), exemples of reconstruction methods include : Auto-encoders, VQ-auto-encoders, PCA, k-means, **specific** versions of GANs, and more recently diffusion models. I would suggest adding the diverse reconstruction methods in the literature review and especially the autoencoders methods, as they are "deep" and can be used for feature extraction (with the encoder) and for reconstruction (encoder+decoder)
- "The feature extractor for these methods is trained on a supervised dataset" if the authors means that the dataset is supervised in the sense that it is cleaned to have no anomalies, then this is the same for every other AD methods mentioned, if not can the authors please explain what they meant ? I would suggest removing the term "supervised".
- The authors state that "training data is created by annotating cells in multiple field-of-views with all normal", but in the end only one field of view is used (40x magnification) ? Can the authors please clarify what they mean.
- In the benchmarking of feature extractor, it seems very clear that the larger the model, the better the performance, could the author provide a graph of performance VS model size ? Or comment this matter ?
- It was not totally clear to me if for the benchmarking task the test set was only composed of dosed rats, meaning that the "normal" patches would be healthy regions of the dosed rats, could the authors emphasize this a little bit ?
- Can the authors explain how the produce the green circles in the Figure 4 ? From what I understand their proposed method is a patch-based method (so it computes squares)
- Another ViT is used for the toxicological study, that is not benchmarked before, why is it the case ? Can the author add this ViT to the benchmark ?

**Special Issue:**

No

---

> ### Author Response · Authors · 2025-03-08
> **Part -1**
>
> We would like to express our sincere appreciation for reviewer’s efforts to review our manuscript. We value the feedback provided by the reviewer, based on which the manuscript has been updated. Below, we address each of the concerns shared by the reviewer.
>
> >"Additionally, (Cai et al., 2024; Bercea et al., 2023) found that AutoDDPM outperformed all other reconstruction-based methods." Please introduce briefly AutoDDPM if mentioned in the literature"
>
> Thank you for the suggestion. We have added a brief description of each method used for the comparative analyses in section 3.2.
>
> > "other methods can be used (GMM, OC-SVM) so I would add that some other methods exist (I feel like the sentence make it seems like only these methods exist)."
>
> Section 1.1 on related work has been revised and improved based on the feedback provided.
>
> > "In the "toxicological study", the training set was reduced from 14 WSI to 4, does this reduction affect the performances a lot ? The authors could provide the AUC of the best performing feature extractor when using only 4 WSI (as I understand the experiments are pretty fast to run due to the method using no training at all)"
>
> The in-distribution dataset was created with the assistance of a pathologist, who identified field-of-views containing normal cells from four control WSIs. This process resulted in a dataset of approximately 2 million normal cells. Due to the large volume of normal cell samples, adding more samples would not significantly affect the performance scores.
>
> > "Could the authors motivate the use of another metric than the AUC in the toxicological study ?"
>
> AUC (Area Under the Curve) was chosen as the evaluation metric due to its widespread use in previous studies [1, 2, 3, 4, 5]. The advantage of using AUC is that it is not a function of a threshold, that is, it does not evaluate the model performance at a particular choice of threshold. AUC provides a more robust and reliable measure of model performance by evaluating the model's ability to distinguish between the two classes across all possible thresholds, offering a more comprehensive view of its overall effectiveness.
>
> > "The authors make then comment that their p-value is above 0.05, while giving the exact value. I would suggest removing the statement about the value being under 0.05"
>
> We appreciate the reviewer for pointing this out. We have made updates to Section 4 and Figure 3 accordingly.
>
> >"I would suggest, to enforce the clarity of the experiments, to use terms like "healthy patches" or "anomalous patches", as it is indeed what's fed to the model and not "cells" "
>
> We have made appropriate changes to the manuscript, for example, we refer to the dataset as ‘cell patch dataset’ instead of ‘cellular dataset’.
>
> >"We compared using average of K-Nearest Neighbour distance and distance to Kth-nearest neighbor, using the later reduced the performance by 0.31%." Can the authors precise what performance they are talking about ? And maybe give the exact value.
>
> We thank the reviewer for bringing this to our attention. We have made the necessary updates to Appendix D.1.
>
> [1] Out-of-distribution detection with deep nearest neighbors [2] Adapting pretrained features for anomaly detection and segmentation [3] A comparative study of anomaly detection in medical images [4] Diffusion models for out-of-distribution detection in digital pathology [5] Exploring the Limits of Out-of-Distribution Detection [6] Rethinking Out-of-distribution (OOD) Detection: Masked Image Modeling is All You Need

---

> > ### Comment · Reviewer_SSPF · 2025-03-11
> >
> > The reviewer would like to thank the authors for taking the time to answer to the comments and for providing a modified version of the manuscript.
> >
> > ```Thank you for the suggestion. We have added a brief description of each method used for the comparative analyses in section 3.2. ```
> > Thank you for the addition.
> >
> > ```Section 1.1 on related work has been revised and improved based on the feedback provided. ```
> > Thank you for the addition.
> >
> > ```The in-distribution dataset was created with the assistance of a pathologist, who identified field-of-views containing normal cells from four control WSIs. This process resulted in a dataset of approximately 2 million normal cells. Due to the large volume of normal cell samples, adding more samples would not significantly affect the performance scores. ```
> > Thank you for the clarification, could the authors state this interesting order of magnitude somewhere in the text (if not already stated) ?
> >
> > ```AUC (Area Under the Curve) was chosen as the evaluation metric due to its widespread use in previous studies [1, 2, 3, 4, 5]. The advantage of using AUC is that it is not a function of a threshold, that is, it does not evaluate the model performance at a particular choice of threshold. AUC provides a more robust and reliable measure of model performance by evaluating the model's ability to distinguish between the two classes across all possible thresholds, offering a more comprehensive view of its overall effectiveness. ```
> > The reviewer feels like the authors did not fully understand the question, I will have it stated more clearly : In section 3 the authors use AUROC as the performance metric. In section 4, unless the reviewer is mistaken, the AUROC could also be used. In section 4, the AUROC is not used, why is that the case ?
> >
> > ```We appreciate the reviewer for pointing this out. We have made updates to Section 4 and Figure 3 accordingly. ```
> > Thank you for the modification.
> > ```We have made appropriate changes to the manuscript, for example, we refer to the dataset as ‘cell patch dataset’ instead of ‘cellular dataset’. ```
> > Thank you for the clarification.
> >
> > ```We thank the reviewer for bringing this to our attention. We have made the necessary updates to Appendix D.1. ```
> > Thank you for the clarification.
> >
> > ```We have updated the introduction section to better explain this, also providing the size of the cell patches in micrometer, for the reference of the readers. ```
> > Thank you for the addition.
> >
> > ```Based on the suggestion, we have updated the related work section 1.2```
> > Thanks for the revision. I still do not quite agree with the way the anomaly detection methods are presented. When talking about “projection methods”, the authors are talking about methods that estimate support/density/distance to the normal distribution, and these methods, most of the time, indeed make use of a projection method first, but the core of the method is support/density estimation or distance estimation. See for instance the very comprehensive review by Ruff et al. on anomaly detection.
> >
> > ```We have added auto-encoders in the related works section 1.2, also a better description of diffusion (DDPMs) are provided in section 1.2 and 3.3```
> > Thanks for the modification.
> >
> > ```We have replaced 'supervised dataset' with 'labeled class dataset', in section 1.2```
> > Thanks for the clarification
> >
> > ```Thank you for highlighting the lack of clarity in the sentence. We acknowledge the oversight and would like to clarify that a pathologist annotated cells across multiple field-of-views from four WSIs to construct the training dataset. This clarification has been made in the updated manuscript. ```
> > Thanks for the clarification
> >
> > ```We have made a similar observation in Section 3.2. Additionally, we found that the model's performance is influenced by both the size of the training dataset and the magnification level of the patch data used during training. ```
> > Thanks for the addition.
> >
> > ```Thank you for your question. The test set contains only healthy and anomalous cell patches from the treated dosage groups. ```
> > Thanks for the clarification, has this been added clearly in the text ?
> >
> > ```For improved visualization, a circular overlay has been applied around the centroid of the cells. This detail has been added to the manuscript in Section Appendix D5. ```
> > The reviewer still does not understand why the authors would create circular overlays where nothing in the method is circular. The visualization would gain clarity and honesty if the patches were represented.

---

> > > ### Comment · Reviewer_SSPF · 2025-03-11
> > >
> > > ```We would like to clarify that Virchow2 is used as the feature extractor for the toxicological study analysis, as it was found to be the best-performing foundation model for the KNN-distance-based method. This has been highlighted in Section 4 of the revised manuscript. ```
> > > Thanks for this clarification.
> > >
> > > Additional typo: “normala”, “observed”, “for cellular anomaly detection, that uses the
> > > largest amount of data, extracting tiles at multiple magnifications. Appendix”
> > >
> > > The reviewer would like to change its rating to weak accept if some of the proposed changes are made, despite the method not being novel (as also pointed out by other reviewers) and evaluated on a private database.

---

> > > > ### Author Response · Authors · 2025-03-12
> > > >
> > > > We thank the reviewer for taking the time to respond to our comments and giving us the opportunity to provide further clarification. We would be happy to make the suggested changes to the manuscript.
> > > >
> > > > >Thank you for the clarification, could the authors state this interesting order of magnitude somewhere in the text (if not already stated) ?
> > > >
> > > > Thank you. We have provided the cell patch count used for the InDistribution dataset in Table 1 of the manuscript.
> > > >
> > > > >In section 3 the authors use AUROC as the performance metric. In section 4, unless the reviewer is mistaken, the AUROC could also be used. In section 4, the AUROC is not used, why is that the case ?
> > > >
> > > > The study-level analysis was conducted on tissue samples from 10 control and 10 treated dosage group animals. However, annotating all anomalous cells across 20 WSIs was not feasible due to the high cell count. Since a labeled dataset was not available for these WSIs, AUROC could not be computed. Nevertheless, box plot analysis revealed that a higher number of cells in the treated group deviated from normal cell patch representations within the feature embedding space compared to the control group. This observation was validated by an expert pathologist, confirming drug-induced changes. We will explicitly mention this in Section 4 for better clarity to the readers.
> > > >
> > > > >Thanks for the revision. I still do not quite agree with the way the anomaly detection methods are presented. When talking about “projection methods”, the authors are talking about methods that estimate support/density/distance to the normal distribution, and these methods, most of the time, indeed make use of a projection method first, but the core of the method is support/density estimation or distance estimation. See for instance the very comprehensive review by Ruff et al. on anomaly detection.
> > > >
> > > > Thank you. Based on the feedback, we will further refine the Related Work section to categorize the approaches based on support/density estimation and distance estimation.
> > > >
> > > > >The reviewer still does not understand why the authors would create circular overlays where nothing in the method is circular. The visualization would gain clarity and honesty if the patches were represented.
> > > >
> > > > We understand the reviewer's concern, we shall be happy to change the figure to show the bounding box corresponding to the patches.
> > > >
> > > > >Additional typo: “normala”, “observed”, “for cellular anomaly detection, that uses the largest amount of data, extracting tiles at multiple magnifications. Appendix”
> > > >
> > > > Thank you again for pointing these out; we shall fix the remaining typos and do another proofreading.
> > > >
> > > > We have requested the Area Chair to allow us to upload the manuscript highlighting the changes from the rebuttle period.

---

> > > > > ### Comment · Reviewer_SSPF · 2025-03-13
> > > > >
> > > > > Thank you for taking the time to respond to these comments.
> > > > >
> > > > > ```The study-level analysis was conducted on tissue samples from 10 control and 10 treated dosage group animals. However, annotating all anomalous cells across 20 WSIs was not feasible due to the high cell count. Since a labeled dataset was not available for these WSIs, AUROC could not be computed. Nevertheless, box plot analysis revealed that a higher number of cells in the treated group deviated from normal cell patch representations within the feature embedding space compared to the control group. This observation was validated by an expert pathologist, confirming drug-induced changes. We will explicitly mention this in Section 4 for better clarity to the readers.```
> > > > > Thank you for this clarification.
> > > > >
> > > > > I would suggest the author to use an external service for uploading the revised manuscript ("dropbox-like"), and to link it in a comment.

---

> > ### Author Response · Authors · 2025-03-13
> >
> > We thank the reviewer for the suggestion.
> >
> > We have uploaded the manuscript with changes highlighted in red for the convenience of reviewers, including the feedback received during the discussion period (section 1.1, section 4 & figure 4), at the following link:
> > https://drive.google.com/file/d/1UvkEzazYMbEPI3msp3P11bjG2LwtuHNu/view?usp=sharing

---

> > > ### Comment · Reviewer_SSPF · 2025-03-14
> > >
> > > I have read the provided revised manuscript and have decided to change my rating to weak accept, as explained in the comments above.
> > > Thanks to the authors for their work and discussion.

---

> ### Author Response · Authors · 2025-03-08
> **Part -2**
>
> > The authors often mention that their study is about "cellular-level" analysis or that they perform "cellular anomaly detection", could the authors precise what they mean by "cellular" in the introduction ? Is it relative to the scale at which the images are magnified ? This term might not be obvious for people not working on WSI.
>
> We have updated the introduction section to better explain this, also providing the size of the cell patches in micrometer, for the reference of the readers.
>
> > " I would suggest being more nuanced in the classification of the AD methods, as they are numerous ways to classify them."
>
> Based on the suggestion, we have updated the related work section 1.2
>
> > "I would suggest adding the diverse reconstruction methods in the literature review and especially the autoencoders methods, as they are "deep" and can be used for feature extraction (with the encoder) and for reconstruction (encoder+decoder)"
>
> We have added auto-encoders in the related works section 1.2, also a better description of diffusion (DDPMs) are provided in section 1.2 and 3.3
>
> >"The feature extractor for these methods is trained on a supervised dataset" if the authors means that the dataset is supervised in the sense that it is cleaned to have no anomalies, then this is the same for every other AD methods mentioned, if not can the authors please explain what they meant ? I would suggest removing the term "supervised".
>
> We have replaced 'supervised dataset' with 'labeled class dataset', in section 1.2
>
> > The authors state that "training data is created by annotating cells in multiple field-of-views with all normal", but in the end only one field of view is used (40x magnification) ? Can the authors please clarify what they mean.
>
> Thank you for highlighting the lack of clarity in the sentence. We acknowledge the oversight and would like to clarify that a pathologist annotated cells across multiple field-of-views from four WSIs to construct the training dataset. This clarification has been made in the updated manuscript.
>
> > "In the benchmarking of feature extractor, it seems very clear that the larger the model, the better the performance, could the author provide a graph of performance VS model size ? Or comment this matter ?"
>
> We have made a similar observation in Section 3.2. Additionally, we found that the model's performance is influenced by both the size of the training dataset and the magnification level of the patch data used during training.
>
> > "It was not totally clear to me if for the benchmarking task the test set was only composed of dosed rats, meaning that the "normal" patches would be healthy regions of the dosed rats, could the authors emphasize this a little bit ?"
>
> Thank you for your question. The test set contains only healthy and anomalous cell patches from the treated dosage groups.
>
> > "Can the authors explain how the produce the green circles in the Figure 4 ? From what I understand their proposed method is a patch-based method (so it computes squares)"
>
> For improved visualization, a circular overlay has been applied around the centroid of the cells. This detail has been added to the manuscript in Section Appendix D5.
>
> > "Another ViT is used for the toxicological study, that is not benchmarked before, why is it the case ? Can the author add this ViT to the benchmark ?"
>
> We would like to clarify that Virchow2 is used as the feature extractor for the toxicological study analysis, as it was found to be the best-performing foundation model for the KNN-distance-based method. This has been highlighted in Section 4 of the revised manuscript.
>
> We would like to sincerely thank the reviewer for taking the time to read our manuscript and our response.

---

> ### Comment · Area_Chair_B326 · 2025-03-13
> **Please update your final rating**
>
> Dear reviewer, the MIDL discussion stage will be end by tomorrow March, 14, 2025. Please read the author's rebuttal and update your final rating. Thanks

---

### Official Review · Reviewer_6oFx · 2025-02-21

**Confidence:** 5
**Preliminary Rating:** 3
**Recommendation:** Poster
**Final Rating:** 4

**Summary:**

The authors introduce an unsupervised method for detecting anomalous cells in toxicological histopathology tissues. They make use of  existing vision encoders pretrained on histopathological datasets as feature extractors and compute anomaly scores using a KNN distance-based algorithm to identify anomalous cell instances. The approach is evaluated using kidney and liver histopathological samples from rats. Threshold-independent metrics (AUC here) show improvement over existing methods.

**Strengths:**

- This is of course an interesting approach for AD in histopathological samples as it addresses some of the main challenges in the field (and medical imaging in general): it is challenging to build large datasets required for supervised methods, and anomalies are typically hard to find and to annotate by experts. Going for an unsupervised approach that requires only nominal cells to build the reference KNN space for anomaly scoring has practical merits for the field.
- The test set has been designed carefully, notably avoiding leakage of tissue slices, and trying to get a balanced nominal-anomalous ratio (which would greatly affect, for instance, threshold-dependent metrics).

**Weaknesses:**

- The novelty of this approach can be argued. Although I do find it valuable to focus on unsupervised (only based on nominal samples to setup a reference feature space) considering the domain of histopathology, the main contribution involves combining existing encoders (as feature extractors) with a simple KNN algorithm for anomaly scoring. More exploration could have been done on the anomaly scoring part in the feature embeddings space.
- Evaluation is limited to only two types of tissues from the same species. This makes it hard to benchmark feature extractors, since performance may be very variable depending on tissue types/species. I would have liked to see a more extensive evaluation by benchmarking on other potential histopathology datasets (from the authors or from other works).
- Also related to the previous point, it would have been more interesting to show performance results by type of anomaly (e.g. necrosis, mitosis, ...). This is assuming the authors have the anomaly type annotations for every cell patch.

**Detailed Comments:**

- What is the typical anomaly rate in the use case of toxicological histopathology? I would assume this can change drastically depending on the drug that is being tested, but it could have been nice to give an estimate or range depending on previous work.
- In the literature review, you don't mention any VAE approaches for reconstruction-based AD methods.
- Check spelling  and writing (e.g. first paragraph of page 5). And more generally, pay more attention to sentence structure (needs more proof reading).
- Future work in conclusion is a bit contradictory: your motivation was to use unsupervised methods to avoid needing data, but you mention that you plan to pretrain feature extractors on larger and more diverse datasets in the future. This seems to go against the idea of addressing the scarcity of samples (and especially anomalous ones) by relying on strong existing feature extractors.
- Some figures in appendix need more work: fig. 6 legend is not clear enough. It is explained in the text, but a figure legend should be self-explanatory without needing to go back to the main text.
- It could have been interesting to visualize the features embedding space via t-sne or UMAP to get more insights on the distribution of nominal vs anomalous feature tensors.

**Justification Of The Final Rating:**

Although I am still not convinced about the novelty of the method, I am willing to increase my score to a weak accept, because I appreciate the authors' efforts to extend the evaluation by providing results of the AUC by anomaly type. Some minor changes such as the literature review are also appreciated.

**Justification Of The Preliminary Rating:**

I see it as borderline, because the approach has merits in the fact that it focused on unsupervised feature extractors and does not require anomalous samples to build the reference feature embeddings, but the evaluation is limited. I may increase my score if the authors extend the evaluation section by providing more fine-grained analysis (e.g. anomaly detection rate by anomaly type) and/or include other datasets for benchmarking.

**Questions To Address In The Rebuttal:**

- If possible, add a more fine-grained evaluation based on the anomaly type. It would be valuable to see if a set of anomaly types as harder to detect (it could also provide insights on what information can the feature extractors capture more effectively).

---

> ### Author Response · Authors · 2025-03-08
>
> We thank the reviewer for the detailed review of the manuscript and for providing valuable feedback. We have updated the manuscript with the feedback. In this rebuttal, we aim to address the concerns raised by the reviewers.
>
> >If possible, add a more fine-grained evaluation based on the anomaly type. It would be valuable to see if a set of anomaly types as harder to detect (it could also provide insights on what information can the feature extractors capture more effectively).
>
> We acknowledge that having class-wise AUC scores would add more value to the analysis. We annotate a dataset for 3 classes for liver tissue  and 1 class for kidney tissue. The fine-grained results are provided in appendix section C.
>
> >The novelty of this approach can be argued. Although I do find it valuable to focus on unsupervised (only based on nominal samples to setup a reference feature space) considering the domain of histopathology, the main contribution involves combining existing encoders (as feature extractors) with a simple KNN algorithm for anomaly scoring.
>
> We found that state-of-the-art diffusion models (DDPM) struggled to differentiate cellular anomalies from normal cells, as the differences are often subtle. While the KNN-distance-based anomaly detection algorithm has been explored in previous studies using ImageNet-pretrained weights or fine-tuning with labeled in-distribution data, this approach does not perform well in unsupervised anomaly detection settings. To the best of our knowledge, no prior work has addressed the detection of subtle cellular anomalies in histopathology image analysis. We used the KNN-distance-based method as a baseline and observed that incorporating foundation models trained on large-scale unlabeled data enabled our method to outperform state-of-the-art DDPMs.
>
> > Future work in conclusion is a bit contradictory: your motivation was to use unsupervised methods to avoid needing data, but you mention that you plan to pretrain feature extractors on larger and more diverse datasets in the future. This seems to go against the idea of addressing the scarcity of samples (and especially anomalous ones) by relying on strong existing feature extractors.
>
> Our goal is to train a self-supervised foundation model on a large-scale cellular dataset that includes samples from multiple organs, dosage groups, toxicological studies, and laboratories. This approach will not require labeled data for any specific anomalous class, but will instead leverage a broad range of unlabeled datasets to train a robust feature extractor for cellular analysis.
>
> > What is the typical anomaly rate in the use case of toxicological histopathology? I would assume this can change drastically depending on the drug that is being tested, but it could have been nice to give an estimate or range depending on previous work.
>
> As mentioned by the reviewer, the percentage of anomalous cells in the tissue varies widely based on the tissue type and drug administered. However, we verified the results of the algorithm with an expert pathologist, who agreed with the findings presented in the paper.
>
> Additionally, we have made the following modifications to the manuscripts
> 1. Added VAE in the literature review
> 2. Added better explanations to figure 1, 3, 4, 7
>
> We thank the reviewer for taking the time to read our response.

---

> ### Comment · Area_Chair_B326 · 2025-03-13
> **Please update your final rating**
>
> Dear reviewer, the MIDL discussion stage will be end by tomorrow March, 14, 2025. Please read the author's rebuttal and update your final rating. Thanks

---

### Official Review · Reviewer_fcFj · 2025-02-21

**Confidence:** 4
**Preliminary Rating:** 3
**Final Rating:** 3

**Summary:**

The paper addresses the challenge of detecting subtle cellular anomalies in toxicological histopathology without requiring labeled anomaly data. It asks how unsupervised anomaly detection can be applied at the cellular level by leveraging foundation models pre-trained on large-scale histopathology data to extract robust feature embeddings. The design employs a KNN-distance–based scoring function to quantify how far a test cell is from its nearest neighbors in the in-distribution feature space, thereby identifying anomalies. This approach aims to distinguish near-out-of-distribution (subtle) cellular variations in both liver and kidney tissues and to evaluate its effectiveness in a toxicological study by comparing control and drug-treated groups.

**Strengths:**

1. The proposed method, which uses a KNN-distance–based anomaly score with a feature extractor pre-trained on histopathology data, outperforms state-of-the-art reconstruction- and projection-based methods.
2. The method is effective in detecting subtle cellular anomalies (near-OOD) that are hard to distinguish from normal cells.
3. Leveraging foundation models pre-trained on large-scale, unsupervised histopathology data improves anomaly detection performance.

**Weaknesses:**

1. The core method of using KNN-distance–based anomaly detection is a well-established approach [1,2,3]. While the paper demonstrates improved performance by leveraging domain-specific foundation models, the algorithmic contribution is relatively incremental compared to existing projection- and reconstruction-based methods.
2. The method relies on a fixed threshold (Q3 + 1.5×IQR) for classifying anomalies, which may be sensitive to variations in the in-distribution data. A more detailed analysis of threshold sensitivity and the potential for adaptive thresholding would strengthen the evaluation.

[1] Automatic medical image classification and abnormality detection using k-nearest neighbour
[2] Progress of machine vision in the detection of cancer cells in histopathology
[3] Visualizing histopathologic deep learning classification and anomaly detection using nonlinear feature space dimensionality reduction

**Detailed Comments:**

1. The introduction is too long and makes it difficult to follow the motivation, the challenges in the field, and the solution provided by the paper. Consider separating the introduction into distinct sections for motivation and related work.
2. It would be beneficial to include an anomaly detection baseline with different backbones, or to implement the method on the same or similar backbones for a more controlled comparison.

**Justification Of The Final Rating:**

I appreciate all the external clarification provided by the author during the rebuttal. However, this study has several limitations: (1) the method lacks innovation in exploring the optimal settings for well-established KNN-based methods in an unsupervised anomaly detection paradigm; and (2) the benchmarking is conducted on a single private dataset, which might not demonstrate the generalizability of the empirical findings. Nonetheless, the benchmarking results could also be of interest to the community, as they illustrate the capabilities of current pathological foundation models in detecting cell abnormalities. Therefore, I would maintain the score as borderline and leave the final decision to my colleagues and the AC.

**Justification Of The Preliminary Rating:**

The paper demonstrates strong performance improvements in unsupervised cellular anomaly detection by leveraging foundation models and a KNN-distance–based scoring function. However, the algorithmic novelty is somewhat incremental given the established nature of KNN-based methods, and the reliance on a fixed threshold raises concerns about robustness.

**Questions To Address In The Rebuttal:**

Please address the weaknesses and detailed comments in your rebuttal.

---

> ### Author Response · Authors · 2025-03-08
>
> We thank the reviewer for taking the time to review our manuscript and for recognizing the importance of our work. Based on the valuable feedback, we have made changes to the manuscript. Below, we would like to address some of the concerns shared by the reviewer.
>
> >The core method of using KNN-distance–based anomaly detection is a well-established approach [1,2,3]. While the paper >demonstrates improved performance by leveraging domain-specific foundation models, the algorithmic contribution is relatively >incremental compared to existing projection- and reconstruction-based methods.
>
> Thank you for sharing the additional references. We agree that KNN-distance-based methods are well-established and have been widely used in previous works, including [1, 2], which we also cite in our study. However, these prior works did not evaluate the KNN-distance method thoroughly and often relied on ImageNet pre-trained feature extractors, which we found to be suboptimal for our task.
>
> Moreover, other studies [1, 6] used feature extractors trained on labeled class data, which is often impractical in the context of unsupervised anomaly detection. In contrast, we demonstrate that using foundation models pre-trained on large-scale datasets as feature extractors significantly improves anomaly detection performance. To the best of our knowledge, this approach has not been explored in previous research. This innovation enabled our method to outperform state-of-the-art diffusion models [3, 4], showcasing its potential for advancing anomaly detection in unsupervised settings.
>
> >The method relies on a fixed threshold (Q3 + 1.5×IQR) for classifying anomalies, which may be sensitive to variations in the in->distribution data. A more detailed analysis of threshold sensitivity and the potential for adaptive thresholding would strengthen the >evaluation.
>
> We thank the reviewer for the valuable feedback. We would like to clarify that Tables 2 and 3 of the manuscript report AUC scores, which are commonly used in prior work [1, 2, 3, 4, 5]. These AUC scores are directly calculated based on the KNN-distance scores,  no thresholding was applied for the comparative analysis.
> To analyze the proportion of anomalous cells at the toxicological study level, we created box plots, as labels are unavailable for the millions of individual cells. A threshold was applied to classify each cell as either normal or anomalous to generate these plots. We used the interquartile range (IQR) method to statistically remove outliers from the distribution of data to derive this threshold. We acknowledge that adjusting the threshold could change the proportion of cells identified as anomalous. However, the box plot in Figure 3 shows that a higher percentage of cells in the treated group tissue exceed this threshold compared to the control group, indicating that more cells in the treated group are significantly distant from normal cellular representations in the feature embedding space. If we generate the box plot with varying thresholds, for example, mean + 3 times the standard deviation, a similar trend is observed. The manuscript has been updated to explain this better.
>
> >The introduction is too long and makes it difficult to follow the motivation, the challenges in the field, and the solution provided by >the paper. Consider separating the introduction into distinct sections for motivation and related work.
>
> We thank the reviewer for their valuable feedback on the introduction section. We recognize the importance of presenting the motivation, challenges, and proposed solution clearly and concisely to effectively set the context for the paper. In response to the reviewer’s suggestion, we have reorganized the introduction into separate sections that now individually address the related work and the motivation behind our study. This restructuring aims to improve clarity and enhance the flow of the paper.
>
> >It would be beneficial to include an anomaly detection baseline with different backbones, or to implement the method on the >same or similar backbones for a more controlled comparison.
>
> We have included baseline results using ImageNet pre-trained models, specifically DINO ViT-B and ResNet-101, in Table 2.
>
> We thank the reviewer for taking the time to read through our responses.
>
> [1] Out-of-distribution detection with deep nearest neighbors [2] Adapting pretrained features for anomaly detection and segmentation [3] A comparative study of anomaly detection in medical images [4] Diffusion models for out-of-distribution detection in digital pathology [5] Exploring the Limits of Out-of-Distribution Detection [6] Rethinking Out-of-distribution (OOD) Detection: Masked Image Modeling is All You Need

---

> ### Comment · Area_Chair_B326 · 2025-03-13
> **Please update your final rating**
>
> Dear reviewer, the MIDL discussion stage will be end by tomorrow March, 14, 2025. Please read the author's rebuttal and update your final rating. Thanks

---

### Author Rebuttal · Authors · 2025-03-08

**Rebuttal:**

We would like to thank all the reviewers for their time and for providing detailed feedback. It is encouraging that the reviewers found this study valuable. We have updated the manuscript based on the feedback (attached) and clarified on the concerns raised by the reviewers in separate comments.

**Supporting Material:**

/attachment/44f951998b9dc9784b4104224e582d6a7483f4da.pdf

---

> ### Comment · Reviewer_SSPF · 2025-03-11
>
> Thanks to the authors for providing an updated version of the manuscript and answers to the reviewers comments. I think most of the reviewers would have appreciated an upload of the revised manuscript with changes highlighted in red(/magenta/whatever).
> Thanks in advance if the authors can provide such a version.

---

> > ### Author Response · Authors · 2025-03-12
> >
> > We apologize for this. We have requested the Area Chair to allow us to upload the manuscript highlighting the changes from the rebuttle period.
> >
> > Thank you.

---

### Comment · Area_Chair_B326 · 2025-03-08

Dear MIDL Reviewers, the discussion stage (March 8–14, 2025) begins today. The authors' rebuttal has been uploaded to OpenReview, and you are encouraged to engage with them for any necessary clarifications. Your participation in the discussion is greatly appreciated.

---

### Author Response · Authors · 2025-03-12
**Upload another version with red**

We thank the area chair for the time and effort towards enabling a smooth rebuttle and discussion period.

We would like to ask if it would be possible to re-upload the revised manuscript highlighting the changes from the rebuttle. We missed doing the same earlier, however, it would be helpful during the discussion with the reviewers.

Best regards

---

### Meta-Review · Area_Chair_B326 · 2025-03-16

**Recommendation:** Accept (Poster)
**Confidence:** 5

**Metareview:**

This paper received two weak acceptances and one borderline review. While the method may not be highly innovative, the paper addresses an interesting topic—detecting toxicological histopathology. I think it is worth discussing at MIDL.